# A Facile Synthesis and Molecular Characterization of Certain New Anti-Proliferative Indole-Based Chemical Entities

**DOI:** 10.3390/ijms24097862

**Published:** 2023-04-26

**Authors:** Reem I. Al-Wabli, Iman S. Issa, Maha S. Al-mutairi, Aliyah A. Almomen, Mohamed I. Attia

**Affiliations:** Department of Pharmaceutical Chemistry, College of Pharmacy, King Saud University, P.O. Box 2457, Riyadh 11451, Saudi Arabia; iman_issa69@yahoo.com (I.S.I.); malmutbiri@ksu.edu.sa (M.S.A.-m.); alalmomen@ksu.edu.sa (A.A.A.); mattia@ksu.edu.sa (M.I.A.)

**Keywords:** substituted benzyl-1*H*-indole-2-carbohydrazide, antiproliferative, IC50

## Abstract

Cancer cells frequently develop drug resistance, which leads to chemotherapeutic treatment failure. Additionally, chemotherapies are hindered by their high toxicity. Therefore, the development of new chemotherapeutic drugs with improved clinical outcomes and low toxicity is a major priority. Several indole derivatives exhibit distinctive anti-cancer mechanisms which have been associated with various molecular targets. In this study, target compounds **4a**–**q** were obtained through the reaction of substituted benzyl chloride with hydrazine hydrate, which produces benzyl hydrazine. Subsequently, the appropriate substituted benzyl hydrazine was allowed to react with 1*H*-indole-2-carboxylic acid or 5-methoxy-1*H*-indole-2-carboxylic acid using 1-ethyl-3-(3-dimethylaminopropyl)carbodiimide as a coupling agent. All compounds exhibited cytotoxicity in three cell lines, namely, MCF-7, A549, and HCT. Compound **4e** exhibited the highest cytotoxicity, with an average IC50 of 2 µM. Moreover, a flow cytometry study revealed a significantly increased prevalence of Annexin-V and 7-AAD positive cell populations. Several derivatives of **4a**–**q** showed moderate to high cytotoxicity against the tested cell lines, with compound **4e** having the highest cytotoxicity, indicating that it may possess potential apoptosis-inducing capabilities.

## 1. Introduction

Cancer, a major causes of mortality worldwide, affects billions of people annually [1]. Despite significant progress in the development of new anticancer drugs, several drawbacks, including low efficacy, high toxicity and drug resistance, affect the outcomes of treatment [2]. Therefore, many researchers aim to develop new anticancer drugs with superior clinical outcomes and a safer profile.

Heterocyclic compounds have a broad range of biological activities and play important roles in medicinal chemistry. In the field of drug discovery, the indole ring is considered an imperative building block owing to its versatile chemistry and wide range of biological functions, including antimicrobial, antiviral, anticonvulsant, analgesic, anti-inflammatory and anticancer activities [3,4,5,6,7,8,9]. Several commercially available potent anticancer drug molecules, such as the natural vinca alkaloids vinblastine and vincristine, contain the indole ring [10]. Additionally, Cediranib is a potent tyrosine kinases receptor inhibitor of vascular endothelial growth factor (VEGF) [11], and Panobinostat, a histone deacetylase inhibitor belonging to the hydroxamic acid class [12].

A literature review has revealed that the distinctive molecular mechanisms regulating the anticancer properties are associated with various molecular targets. Some of the indole derivatives inhibit tubulin polymerization, and induce apoptosis in cancer cells [13,14,15,16]. However, other derivatives were found that inhibit protein kinases, such as casein kinase 2 (CK2), tyrosine kinases (TrK), and cyclin-dependent kinases 4 and 6 (CDK), as well as vascular endothelial growth factor-2 (VEGFR). Furthermore, indoles can induce apoptosis via the inhibition of Mcl-1, Bcl-2, and poly(ADP-ribose) polymerase 1 (PARP), which prevents DNA repair. It also exhibits anticancer effects by suppressing DNA topoisomerases, aromatases, and histone deacetylases (HDAC) [9,17,18,19,20].

Therefore, compounds containing indole moieties exhibit different mechanisms of action involving multiple targets during cell replication and proliferation. Considering this, and in continuation of ousr previous research on new potent anti-proliferative compounds [21,22], in this study we aimed to synthesize indole backbone molecules tethered with free rotating substituted benzyl moieties **4a**–**q**. The flexibility of the molecules is a modality for obtaining a number of bioactive confirmations, and hence increasing the likelihood of obtaining new bioactive compounds with effective anti-proliferative properties.

## 2. Results and Discussions

### 2.1. Chemistry

The synthetic pathway used to prepare the target compounds **4a**–**q** is depicted in Figure 1. Commercially available substituted benzyl chloride **1** was allowed to react with hydrazine hydrate in ethanol to give benzyl hydrazine **2**. Subsequently, the title compounds **4a**–**q** were achieved by reacting the appropriate substituted benzyl hydrazine with 1*H*-indole-2-carboxylic acid or 5-methoxy-1*H*-indole-2-carboxylic acid in dichloromethane using 1-ethyl-3-(3-dimethylaminopropyl)carbodiimide hydrochloride (EDCI) as a coupling agent. The direct conversion of a carboxylic acid to an amide is difficult because amines are very basic and tend to convert carboxylic acids to their highly unreactive carboxylate ions. Therefore, EDCI as a coupling agent was used to drive this reaction by forming a good leaving group which could then be displaced by an amine during the addition elimination reaction [23]. The nucleophilic attack happened due to the secondary amine rather than the primary amine of the hydrazine terminal, which led to the formation of the sole product in the target compounds **4a**–**q**. The assigned structures were confirmed by ^1^HNMR, ^13^CNMR, ESI-MS, and X-ray crystallography.

The ^1^H-NMR spectra of the target compounds **4a**–**q** showed D_2_O exchangeable singlet peaks integrated for two protons in the range of δ 4.81–5.03 ppm assigned for the NH_2_. Other singlets integrated for two protons in the range of δ 4.91–5.34 ppm were noticed for the benzylic CH_2_. Meanwhile, compounds **4b**, **4d**, **4f**, **4h**, **4j**, **4l**, **4n**, **4p**, and **4q** showed singlets at around δ = 3.76 ppm, representing three protons of the indole -OCH_3_ group. Compound **4q** exhibited a singlet at δ = 2.30 ppm, which was assigned to be for the three protons of the CH_3_ group. The aromatic protons were observed in the region of δ = 6.84–8.25 ppm. Moreover, the indole NH appeared as singlets integrated for one proton in the range δ = 11.36–11.55 ppm, whereas the ^13^C-NMR spectra of the target compounds **4a**–**q** exhibited signals in the range of δ = 54.0–55.6 ppm, indicating carbons of the benzylic CH_2_. The methyl carbons of compound **4q** were observed at δ = 21.2 ppm, and the indole methoxy carbon for compounds **4b**, **4d**, **4f**, **4h**, **4j**, **4l**, **4n**, **4p**, and **4q** resonated at δ = 55.7 ppm. Moreover, the aromatic carbons appeared in the range of δ = 102.4–158.9 ppm while the amide carbonyl carbon appeared at about δ 163 ppm.

### 2.2. Antiproliferative Activity

The title compounds **4a**–**q** were subjected to MTT assay to determine their cytotoxicity, and their in vitro antiproliferative activity was examined as well. The cancer cell growth inhibitory activity of the synthesized compound was tested against three human cancer cell lines, namely, breast cancer (MCF-7), colon cancer (HCT116) and lung cancer (A549), as well as non-tumorigenic human lung cell line (WI38) to estimate the selectivity for tumor cells. Staurosporine was used as a reference drug.

All the title compounds **4a**–**q** were evaluated for their quantitative inhibitory concentration 50% (IC50), and the results are depicted in Table 1 and Figure 1. It was obvious from the results that several derivatives showed moderate to high cytotoxicities. Compounds **4b**, **4h**, and **4a** showed moderate cytotoxicities, with IC50s 11.5, 13.1, and 21.7 µM, respectively, against MCF-7, while for compounds **4j**, **4f**, **4g**, **4m**, and **4p**, IC50s were 9.16, 9.79, 12.3, 17.5, and 19.4 µM, respectively, against HCT116 and compounds **4i**, **4m**,**4n**, **4g**, and **4h** IC50s; 11.6, 12.9, 16.7, 17.0, and 23.9 µM, respectively, compared to Staurasporine against the same cell line (IC50s: 11.1, 7.02, and 8.42 µM, respectively).

Furthermore, it appeared that compounds **4e**, **4q**, **4d**, **4j**, **4o**, **4g**, and **4k** revealed high cytotoxicities against MCF-7 with IC50s (0.57, 1.01, 3.23, 3.27, 3.66, 8.31, and 9.21 µM, respectively). Additionally, compounds **4e**, **4o**, **4l**, and **4q** with IC50s (1.95, 2.41, 5.02, and 6.45 µM, respectively) were found to be superior to staurasporine against HCT116, while compounds **4q**, **4k**, **4p**, **4e**, **4d**, **4a**, and **4f** displayed significant cytotoxicities against A549 (IC50s, 2.4, 2.65, 3.02, 3.49, 5.69, 5.9, and 8.33 µM, respectively).

Among the tested target compounds, **4e** and **4q** showed excellent cytotoxicity with average IC50s of 2 ± 1.2 and 3.28 ± 2.3 µM, respectively (Table 1, Figure 1). However, compound **4e** exhibited less toxicity on normal cells (WI-38) in comparison to **4q** (Figure 2), 87.2 ± 4.94 and 56.2 ± 3.18, respectively, and thus was chosen for further investigation.

### 2.3. Flowcytometry

Since a paramount goal for anticancer agents is to induce apoptosis and cause malfunctions in the DNA, flowcytometry was conducted in this study for compound **4e** and the results (Figure 3) depicted a significant increase in Annexin-V and 7-AAD positive cell populations, which indicated that compound **4e** harbors potential apoptosis-inducing capabilities.

### 2.4. Cell Cycle Arrest

To evaluate if the antiproliferative effect of **4e** was due to the disturbance of the cell cycle, cells were treated with 10 µm for 48 h. Results show that there was an increase in the cell population in the S phase (48.72%) compared to control (34.72%) in MCF-7 cells, about a 1.04-fold increase (Figure 4). Therefore, it is possible that the antiproliferative effect of **4e** is due to S-phase cell cycle arrest.

## 3. Materials and Methods

### 3.1. General

The melting points were measured using a Gallenkamp melting point device and are uncorrected. The NMR samples of the synthesized compounds **4a**–**q** were dissolved in DMSO-*d*_6_, and the NMR spectra were recorded using a Bruker NMR spectrometer (Bruker, Reinstetten, Germany) at 500/700 MHz for ^1^H and 125.76/175 MHz for ^13^C at the Research Center, College of Pharmacy, King Saud University, Saudi Arabia. TMS was used as an internal standard, and chemical shift values were recorded in ppm on the δ scale. The ^1^H NMR spectral data are represented as follows: chemical shifts, multiplicity (s, singlet; d, doublet; t, triplet; and m, multiplet), and number of protons. The ^13^C NMR spectral data were represented as chemical shifts and type of carbon. Mass spectra were measured on an Agilent Quadrupole 6120 LC/MS with ESI (electrospray ionization) source (Agilent Technologies, Palo Alto, CA, USA). Elemental analysis was carried out at the Microanalysis Laboratory, Cairo University, Cairo, Egypt, using an Elemental C, H, N analyzer Vario EL III, Germany, and the results agreed favorably with the proposed structures within ±0.4% of the theoretical values. Silica gel thin layer chromatography (TLC) plates from Merck, Burlington, MA, USA (silica gel precoated aluminum plates with fluorescent indicator at 254 nm) were used for thin layer chromatography. Visualization was performed by illumination with a UV light source (254 nm). Cell line cells were purchased from American Type Culture Collection, and cells were cultured using DMEM (Invitrogen/Life Technologies, Carlsbad, CA, USA) supplemented with 10% Hyclone FBS, 10 ug/mL of insulin (Sigma, St. Louis, MO, USA), and 1% penicillin-streptomycin. All of the other chemicals and reagents were from Sigma or Invitrogen.

### 3.2. Chemistry

#### 3.2.1. General Method for the Synthesis of Substituted Benzyl Hydrazine **3**

The appropriate benzyl chloride (0.01 mol) dissolved in 8 mL absolute ethanol was added dropwise to a stirred solution of 98% hydrazine hydrate (6 mL, 0.12 mol) in 12 mL absolute ethanol, and the resulting mixture continued to be stirred at room temperature for 24 h. Then, the solvent was evaporated under reduced pressure, 0.12 mL of aqueous solution 50% NaOH was added to the residue, and the resulting mixture was extracted by diethyl ether (3 × 20 mL). The combined organic layer was dried over anhydrous Na_2_SO_4_ and then concentrated to afford benzyl hydrazine with 60–80 yield %, which was used in the next step without further purification.

#### 3.2.2. General Method for the Synthesis of the Target Compounds **4a**–**q**

A mixture of indole-2-carboxylic acid 3 (0.25 g, 1.55 mmol) and EDCI (0.3 g, 1.55 mmol) in methylene chloride (5 mL) was stirred for about 10 min until a clear solution was obtained. Substituted benzyl hydrazine 2 (1.55 mmol) dissolved in methylene chloride (5 mL) was added to the mixture and stirred for 24 h. The mixture was washed successively with water (2 × 20 mL), 10% NaHCO_3_ solution (2 × 15 mL), and water (2 × 15 mL). The combined organic layer was dried over anhydrous Na_2_SO_4_ and concentrated. The formed precipitate was collected and re-crystallized from ethanol to yield the corresponding **4a**–**q**.

*N-(4-Chlorobenzyl)-1H-indole-2-carbohydrazide* **4a**: beige powder m.p. 233 °C (yield; 0.26 g, 56%); IR (KBr): ν (cm^−1^) 3348 (NH_2asym_.), 3280 (NH_2sym._), 3061 (C-H, aromatic), 2950 (C-H, aliphatic), 1701 (C=O), 1560, 1163, 748; ^1^H NMR (500 MHz, DMSO-*d*_6_) *ppm*: 4.85 (s, 2H, NH_2_), 5.06 (s, 2H, CH_2_), 7.01–7.04 (m, 1H, Har), 7.17–7.20 (m, 1H, Har), 7.37 (d, 2H, *J* = 8.5 Hz, Har), 7.44 (d, 2H, *J* = 8.5 Hz, Har), 7.49 (d, 2H, *J* = 8 Hz, Har), 7.61 (d, 1H, *J* = 7.5 Hz, Har), 11.50 (s, 1H, NH-indole); ^13^C NMR (125 MHz, DMSO-*d*_6_) *ppm*: 54.0 (CH_2_), 112.8, 120.0, 122.1, 123.9, 127.4, 128.9, 130.4, 132.3, 136.3, 136.5, 162.7 (C=O); MS *m*/*z* (ESI): 297.8 [M − H]^+^; anal. calcd. For C_16_H_14_ClN_3_O: C, 64.11; H, 4.71; N, 14.02; found: C, 64.35; H, 4.70; N, 13.99.

*N-(4-Chlorobenzyl)-5-methoxy-1H-indole-2-carbohydrazide* **4b**: white powder m.p. 225 °C (yield; 0.37 g, 86%); IR (KBr): ν (cm^−1^) 3315 (NH_2asym._), 3250 (NH_2sym._), 3100 (C-H, aromatic), 2970 (C-H, aliphatic), 1690 (C=O), 1585, 796; ^1^H NMR (700 MHz, DMSO-*d*_6_) *ppm*: 3.75 (s, 3H, OCH_3_), 4.84 (s, 2H, NH_2_), 5.04 (s, 2H, CH_2_), 6.85 (m, 1H, Har), 7.07 (s, 1H, Har), 7.37 (m, 4H, Har), 7.44 (d, 2H, *J* = 8.5 Hz, Har), 11.37 (s, 1H, NH-indole); ^13^C NMR (175 MHz, DMSO-*d*_6_) *ppm*: 54.0 (CH_2_), 55.6 (OCH_3_), 102.2, 108.9, 113.5, 115.4, 127.6, 128.9, 130.4, 131.0, 131.7, 132.3, 136.5, 154.2 (C-O), 162.6 (C=O); MS *m*/*z* (ESI): 327.8 [M − H]^+^; anal. calcd. for C_17_H_16_ClN_3_O_2_: C, 61.91; H, 4.89; N, 12.74; found: C, 61.67; H, 4.90; N, 12.73.

*N-Benzyl-1H-indole-2-carbohydrazide* **4c**: white powder m.p. 237 °C (yield; 0.144 g, 35 %); IR (KBr): ν (cm^−1^) 3313 (NH_2asym_.), 3280 (NH_2sym._), 3100 (C-H, aromatic), 2980 (C-H, aliphatic), 1680 (C=O), 1585, 744; ^1^H NMR (500 MHz, DMSO-*d*_6_) *ppm*: 4.88 (s, 2H, NH_2_), 5.00 (s, 2H, CH_2_), 7.26 (t, 1H, *J* = 7 Hz, Har), 7.18 (t, 1H, *J* = 7 Hz, Har), 7.31–7.39 (m, 5H, Har), 7.49 (d, 2H, *J* = 8 Hz, Har), 7.62 (d, 1H, *J* = 7 Hz, Har), 11.50 (s, 1H, NH-indole); ^13^C NMR (125 MHz, DMSO-*d*_6_) *ppm*: 54.5 (CH_2_), 112.8, 119.9, 122.1, 123.8, 127.5, 127.7, 128.4, 129.0, 136.3, 137.4, 162.6 (C=O); MS *m*/*z* (ESI): 263.8 [M − H]^+^; anal. calcd. For C_16_H_15_N_3_O: C, 72.43; H, 5.70; N, 15.84; found: C, 72.45; H, 5.67; N, 15.83.

*N-Benzyl-5-methoxy-1H-indole-2-carbohydrazide* **4d**: beige powder m.p. 210 °C (yield; 0.143 g, 37 %); IR (KBr): ν (cm^−1^) 3331 (NH_2asym._), 3290 (NH_2sym._), 3100 (C-H, aromatic), 2950 (C-H, aliphatic), 1670 (C=O), 1585, 744; ^1^H NMR (500 MHz, DMSO-*d*_6_) *ppm*: 3.76 (s, 1H, OCH_3_), 4.81 (s, 2H, NH_2_), 4.91 (s, 2H, CH_2_), 6.83 (dd, 1H, *J* = 2.5, 9Hz, Har), 7.07 (s, 1H, Har), 7.18–7.24 (m, 5H, Har), 7.39 (d, 2H, *J* = 7Hz), 11.63 (s, 1 H, NH-indole); ^13^C NMR (DMSO-*d*_6_) *ppm*: 55.6 (CH_2_), 55.7(OCH_3_), 102.7, 113.7, 110.5, 116.4, 127.8, 128.0, 129.6, 132.3, 135.0, 136.0, 142.7, 154.3, 162.5 (C=O); MS *m*/*z* (ESI): 293.9 [M − H]^+^; anal. calcd. for C_17_H_17_N_3_O_2_: C, 69.14; H, 5.80; N, 14.23; found: C, 69.02; H, 5.82; N, 14.25.

*N-(3,5-Bis(trifluoromethyl)benzyl)-1H-indole-2-carbohydrazide* **4e**: white powder m.p. 204 °C (yield; 0.261g, 42%); IR (KBr): ν (cm^−1^) 3332 (NH_2asym._), 3228 (NH_2sym._), 3100 (C-H, aromatic), 2940 (C-H, aliphatic), 1670 (C=O), 1521, 1178, 1130, 744; ^1^H NMR (700 MHz, DMSO-*d*_6_) *ppm*: 5.02 (s, 2H, CH_2_), 5.34 (s, 2 H, NH_2exchang_.), 7.03 (t, 1 H, *J* = 7 Hz, Har), 7.18–7.19 (t, 1 H, Har), 7.49 (2 H, *J* = 7 Hz, Har), 7.63 (d, 1 H, *J* = 7 Hz, Har), 8.05 (s, 3 H, Har), 11.55 (s, 1 H, NH-indole-_exchang._); ^13^C NMR (175 MHz, DMSO-*d*_6_) *ppm*: 54.8 (-CH_2_), 109.3, 112.9, 119.9, 121.5, 124.0, 126.2, 127.5, 129.4, 130.3, 130.4, 130.6, 130.8, 131.0, 136.3, 141.3, 163.0 (C=O); MS *m*/*z* (ESI): 399.9 [M − H]^+^; anal. calcd. for C_18_H_13_F_6_N_3_O: C, 53.87; H, 3.27; N, 10.47; found: C, 54.06; H, 3.26; N, 10.49.

*N-(3,5-Bis(trifluoromethyl)benzyl)-5-methoxy-1H-indole-2-carbohydrazide* **4f**: white powder m.p. 195 °C (yield; 0.215 g, 38%); IR (KBr): ν (cm^−1^) 3342 (NH_2asym._), 3200 (NH_2sym._), 3149 (C-H, aromatic), 2937 (C-H, aliphatic), 1640 (C=O), 1521, 1165, 1138, 740; ^1^H NMR (500 MHz, DMSO-*d*_6_) *ppm*: 3.76 (s, 3 H, OCH_3_), 5.01 (s, 2 H, NH_2_), 5.32 (s, 2 H, CH_2_), 6.84–6.87 (m, 1 H, Har), 7.08 (s, 1 H, Har), 7.39 (d, 2 H, *J* = 5 Hz, Har), 8.03 (s, 3 H, Har), 11.43 (s, 1 H, NH-indole); ^13^C NMR (125 MHz, DMSO-*d*_6_) *ppm*: 54.8 (-CH_2_), 55.7 (OCH_3_), 102.4, 113.7, 115.3, 121.5, 122.8, 125.0, 127.6, 129.4, 130.6, 130.8, 131.8, 141.4, 154.1, 163.0(C=O); MS *m*/*z* (ESI): 429.8 [M − H]^+^; anal. calcd. for C_19_H_15_F_6_N_3_O_2_: C, 52.91; H, 3.51; N, 9.74; found: C, 52.71; H, 3.52; N, 9.74.

*N-(4-Fluorobenzyl)-1H-indole-2-carbohydrazide beige powder* **4g**: m.p. 209 °C (yield; 0.228 g, 52%); IR (KBr): ν (cm^−1^) 3315 (NH_2asym._), 3280 (NH_2sym._), 3050 (C-H, aromatic), 2937 (C-H, aliphatic), 1647(C=O), 1517, 1151, 746; ^1^H NMR (700 MHz, DMSO-*d*_6_) *ppm*: 4.85 (s, 2 H, NH_2_), 5.03 (s, 2 H, CH_2_), 7.01–7.04 (t, 1 H, *J* = 7 Hz, Har), 7.17–7.23 (m, 3 H, Har), 7.39–7.41 (m, 2 H, Har), 7.49 (d, 2H, *J* = 7 Hz), 7.61 (d, 1 H, *J* = 8 Hz, Har), 11.49 (s, 1 H, NH-indole); ^13^C NMR (175 MHz, DMSO-*d*_6_) *ppm*: 55.0 (-CH_2_), 109.4, 112.8, 115.7, 115.9, 119.9, 122.0, 123.9, 127.5, 130.6, 133.5, 136.3, 161.0, 162.6 (C=O); MS *m*/*z* (ESI): 281.9 [M − H]^+^; anal. calcd. for C_16_H_14_FN_3_O: C, 67.3; H, 4.98; N, 14.83; found: C, 67.08; H, 4.99; N, 14.86.

*N-(4-Fluorobenzyl)-5-methoxy-1H-indole-2-carbohydrazide* **4h**: beige powder m.p. 175 °C (yield; 0.225 g, 52%); IR (KBr): ν (cm^−1^) 3338 (NH_2asym._), 3209 (NH_2sym._), 3116 (C-H, aromatic), 2964 (C-H, aliphatic), 1635 (C=O), 1527, 1155, 754; ^1^H NMR (500 MHz, DMSO-*d*_6_) *ppm*: 3.76 (s, 3 H, OCH_3_), 4.84 (s, 2 H, NH_2_), 5.00 (s, 2 H, CH_2_), 6.84–6.86 (m, 1 H, Har), 7.07 (s, 1 H, Har), 7.19–7.22 (m, 2 H, Har), 7.38–7.39 (m, 4 H, Har), 11.36 (s, 1 H, NH-indole); ^13^C NMR (125 MHz, DMSO-*d*_6_) *ppm*: 53.9 (-CH_2_), 55.7 (OCH_3_), 102.4, 113.7, 115.1, 115.6, 115.8, 127.7, 130.5, 131.7, 133.7, 154.0, 162.5 (C-O), 163.1 (C=O); MS *m*/*z* (ESI): 311.9 [M − H]^+^; anal. calcd. for C_17_H_16_FN_3_O_2_: C, 65.17; H, 5.15; N, 13.41; found: C, 65.36; H, 5.17; N, 13.36.

*N-(4-(Trifluoromethyl)benzyl)-1H-indole-2-carbohydrazide* **4i**: white powder m.p. 190 °C (yield; 0.255 g, 43%); IR (KBr): ν (cm^−1^) 3334 (NH_2asym._), 3210 (NH_2sym._), 3100 (C-H, aromatic), 2937 (C-H, aliphatic), 1658 (C=O), 1564, 1112, 745; ^1^H NMR (700 MHz, DMSO-*d*_6_) *ppm*: 4.97 (NH_2_), 5.15 (CH_2_), 7.03 (t, 1H, *J* = 7.5 Hz, Har), 7.193 (t, 1H, *J* = 7.5 Hz, Har), 7.50 (d, 2H, *J* = 8 Hz, Har), 7.56 (d, 2H, *J* = 7.5 Hz, Har), 7.62 (d, 1H, *J* = 8 Hz, Har), 7.70 (d, 2H, *J* = 8 Hz, Har), 11.52 (s, 1H, NH-indole); ^13^C NMR (175 MHz, DMSO-*d*_6_) *ppm*: 54.0 (CH_2_), 109.4, 112.8, 119.9, 122.1, 123.9, 124.0, 125.9, 127.5, 128.2, 128.4, 129.1, 130.4, 136.4, 142.4, 162.8 (C=O); MS *m*/*z* (ESI): 331.9 [M − H]^+^; anal. calcd. for C_17_H_14_F_3_N_3_O: C, 61.26; H, 4.23; N, 12.61; found: C, 61.38; H, 4.21; N, 12.60.

*N-(4-(Trifluoromethyl)benzyl)-5-methoxy-1H-indole-2-carbohydrazide* **4j**: white powder m.p. 220 °C (yield; 0.21 g, 45%); IR (KBr): ν (cm^−1^) 3332 (NH_2asym._), 3270 (NH_2sym._), 3050 (C-H, aromatic), 2940 (C-H, aliphatic), 1600 (C=O), 1521, 1112, 805; ^1^H NMR (500 MHz, DMSO-*d*_6_) *ppm*: 3.76 (s, 3 H, OCH_3_), 4.95 (s, 2 H, NH_2_), 5.13 (s, 2 H, CH_2_), 6.85 (dd, 1 H, *J* = 2, 9 Hz, Har), 7.08 (s, 1H, Har), 7.40 (d, 2 H, *J* = 8.5 Hz, Har), 7.56 (d, 2 H, *J* = 7.5 Hz, Har), 7.75 (d, 2 H, *J* = 7.5 Hz, Har), 11.40 (s, 1 H, NH-indole); ^13^C NMR (125 MHz, DMSO-*d*_6_) *ppm*: 54.5 (CH_2_), 55.7 (OCH_3_), 102.4, 113.7, 115.2, 123.8, 125.8, 127.7, 128.2, 128.5, 129.0, 130.9, 131.8, 142.5, 154.1, 162.7 (C=O); MS *m*/*z* (ESI): 36 1.9 [M − H]^+^; anal. calcd. for C_18_H_16_F_3_N_3_O_2_: C, 59.50; H, 4.44; N, 11.57; found: C, 59.42; H, 4.45; N, 11.56.

*N-(4-Cyanobenzyl)-1H-indole-2-carbohydrazide* **4k**: white powder m.p. 202 °C (yield; 0.315 g, 70%); IR (KBr): ν (cm^−1^) 3379 (NH_2asym._), 3311 (NH_2sym._), 3047 (C-H, aromatic), 2922 (C-H, aliphatic), 2223 (CN), 1647 (C=O), 1523, 740; ^1^H NMR (500 MHz, DMSO-*d*_6_) *ppm*: 4.95 (s, 2 H, NH_2_), 5.17 (s, 2 H, CH_2_), 7.01–7.03 (m, 1H, Har), 7.18–7.21 (m, 1H, Har), 7.50 (d, 2H, *J* = 8.5 Hz, Har), 7.53 (d, 2 H, *J* = 8 Hz, Har), 7.62 (d, 1H, *J* = 7.5 Hz, Har), 7.86 (d, 2 H, *J* = 8 Hz, Har), 11.52 (s, 1H, NH-indole);^13^C NMR (125 MHz, DMSO-*d*_6_) *ppm*: 54.8 (CH_2_), 110.4, 112.8, 119.4, 120.0, 122.1, 123.9, 130.4, 132.9, 136.4, 143.6, 162.9 (C=O); MS *m*/*z* (ESI): 288.9 [M − H]^+^; anal. calcd. for C_17_H_14_N_4_O: C, 70.33; H, 4.86; N, 19.30; found: C, 70.05; H, 4.88; N, 19.36.

*N-(4-Cyanobenzyl)-5-methoxy-1H-indole-2-carbohydrazide* **4l**: white powder m.p. 248 °C (yield; 0.311 g, 74%); IR (KBr): ν (cm^−1^) 3390 (NH_2asym._), 3331(NH_2sym._), 3060 (C-H, aromatic), 2972 (C-H, aliphatic), 2227 (CN), 1647 (C=O), 1517, 750; ^1^H NMR (500 MHz, DMSO-*d*_6_) *ppm*: 3.76 (s, 3 H, OCH_3_), 4.94 (s, 2H, NH_2_), 5.16 (s, 2 H, CH_2_), 6.86 (dd, 1H, *J* = 2.5, 9 Hz, Har), 7.08 (s, 1H, Har), 7.40 (d, 2H, *J* = 8.5 Hz, Har), 7.53 (d, 2 H, *J* = 8 Hz, Har), 7.85 (d, 2H, *J* = 8 Hz, Har),11.39 (s, 1H, NH-indole);^13^C NMR (125 MHz, DMSO-*d*_6_) *ppm*: 54.8 (CH_2_), 55.7 (OCH_3_), 102.4, 108.9, 110.4, 113.7, 115.3, 119.4, 127.7, 129.2, 131.8, 132.9, 143.6, 154.1, 162.8 (C=O); MS *m*/*z* (ESI): 318.9 [M − H]^+^; anal. calcd. for C_18_H_16_N_4_O_2_: C, 67.49; H, 5.03; N, 17.49; found: C, 67.28; H, 5.03; N, 17.54.

*N-(4-Bromobenzyl)-1H-indole-2-carbohydrazide beige powder* **4m**: m.p. 238 °C (yield; 0.133 g, 25%); IR (KBr): ν (cm^−1^) 3334 (NH_2asym._), 3210 (NH_2sym._), 3050 (C-H, aromatic), 2960 (C-H, aliphatic), 1637 (C=O), 1521, 788, 750; ^1^H NMR (500 MHz, DMSO-*d*_6_) *ppm*: 4.84 (s, 2 H, NH_2_), 5.05 (s, 2 H, CH_2_), 7.03 (m, 1 H, Har), 7.19 (m, 1H, Har), 7.31 (d, 2H, *J* = 8 Hz, Har), 7.49–7.62 (m, 5H, Har), 11.45 (s, 1H, NH-indole); ^13^C NMR (125 MHz, DMSO-*d*_6_) *ppm*: 54.7 (CH_2_), 112.8, 119.9, 120.8, 122.1, 123.9, 127.5, 130.7, 131.3, 131.5, 131.9, 136.4, 136.9, 162.7 (C=O); MS *m*/*z* (ESI): 341.9 [M − H]^+^; anal. calcd. for C_16_H_14_BrN_3_O: C, 55.83; H, 4.10; N, 12.21; found: C, 55.97; H, 4.11; N, 12.15.

*N-(4-Bromobenzyl)-5-methoxy-1H-indole-2-carbohydrazide* **4n**: beige powder m.p. 228 °C (yield; 0.157 g, 32%); IR (KBr): ν (cm^−1^) 3415 (NH_2asym._), 3331(NH_2sym._), 3050 (C-H, aromatic), 2926 (C-H, aliphatic),1640 (C=O), 1519, 792, 750; ^1^H NMR (500 MHz, DMSO-*d*_6_) *ppm*: 3.76 (s, 3 H, OCH_3_), 4.83 (s, 2 H, NH_2_), 5.04 (s, 2 H, CH_2_), 6.85 (dd, 1 H, *J* = 2.5, 9 Hz, Har), 7.08 (s, 1 H, Har), 7.31 (d, 2 H, *J* = 8 Hz, Har), 7.39 (d, 2H, *J* = 9 Hz, Har), 7.58 (d, 2 H, *J* = 8 Hz, Har), 11.40 (s, 1H, NH-indole); ^13^C NMR (125 MHz, DMSO-*d*_6_) *ppm*: 54.1 (CH_2_), 55.7 (OCH_3_), 102.4, 113.7, 115.2, 120.8, 127.6, 130.8, 131.0, 131.7, 131.9, 137.0, 154.0, 162.6 (C=O); MS *m*/*z* (ESI): 372.8 [M − H]^+^; anal. calcd. for C_17_H_16_BrN_3_O_2_: C, 54.56; H, 4.31; N, 11.23; found: C, 54.55; H, 4.30; N, 11.26.

*N-(4-Nitrobenzyl)-1H-indole-2-carbohydrazide* **4o**: yellow powder m.p. 223 °C (yield; 0.37 g, 77%); IR (KBr): ν (cm^−1^) 3336 (NH_2asym._), 3250 (NH_2sym._), 3070 (C-H, aromatic), 2940 (C-H, aliphatic), 1630 (C=O), 1593, 1517, 1348, 746; ^1^H NMR (700 MHz, DMSO-*d*_6_) *ppm*: 4.99 (s, 2H, NH_2_), 5.19 (s, 2H, CH_2_), 7.03 (t, 1H, *J* = 7.5 Hz, Har), 7.19 (t, 1H, *J* = 7.5 Hz, Har), 7.49 (d, 2H, *J* = 8 Hz, Har), 7.61 (d, 3H, *J* = 8 Hz, Har), 8.25 (d, 2H, *J* = 8.5 Hz, Har), 11.52 (s, 1H, NH-indole); ^13^C NMR (175 MHz, DMSO-*d*_6_) *ppm*: 54.5 (CH_2_), 109.2, 112.8, 120.2, 122.1, 123.9, 124.1, 127.5, 129.7, 130.4, 147.2, 162.9 (C=O); MS *m*/*z* (ESI): 308.9 [M − H]^+^; anal. calcd. for C_16_H_14_N_4_O_3_: C, 61.93; H, 4.55; N, 18.06; found: C, 61.88; H, 4.56; N, 18.13.

*N-(4-Nitrobenzyl)-5-methoxy-1H-indole-2-carbohydrazide* **4p**: yellow powder m.p. 213 °C (yield; 0.285 g, 64%); IR (KBr): ν (cm^−1^) 3394 (NH_2asym._), 3317 (NH_2sym._), 3060 (C-H, aromatic), 2950 (C-H, aliphatic), 1654 (C=O), 1589, 1517, 1348, 750;^1^H NMR (700 Hz, DMSO-*d*_6_) *ppm*: 3.76 (s, 3H, -OCH_3_), 4.99 (s, 2H, NH_2_), 5.18 (s, 2H, CH_2_), 6.85 (dd, 1H, *J* = 2, 9 Hz, Har), 7.08 (s, 1H, Har), 7.39 (d, 2H, *J* = 9Hz, Har), 7.60 (d, 2H, *J* = 8.5 Hz, Har), 8.25 (d, 2H, *J* = 8.5 Hz, Har), 11.40 (s, 1H, NH-indole). ^13^C NMR (175 MHz, DMSO-*d*_6_) *ppm*: 54.1 (CH_2_), 55.7 (-OCH_3_), 102.1, 109.1, 113.7, 115.3, 124.1, 127.6, 129.4, 130.8, 131.8, 147.2, 145.8, 154.1, 162.8 (C=O); MS *m*/*z* (ESI): 338.9 [M − H]^+^; anal. calcd. for C_17_H_16_N_4_O_4_: C, 59.99; H, 4.74; N, 16.46; found: C, 59.94; H, 4.74; N, 16.50.

*N-(4-Methylbenzyl)-5-methoxy-1H-indole-2-carbohydrazide* **4q**: beige powder m.p. 166 °C (yield; 0.122 g, 30%); IR (KBr): ν (cm^−1^) 3309 (NH_2asym_), 3230 (NH_2sym._), 3050 (C-H, aromatic), 2920 (C-H, aliphatic), 1577, 1570 (C=N), 1494, 1450, 750; ^1^H NMR (700 MHz, DMSO-*d*_6_) *ppm*: 2.30 (s, 3H, -CH_3_), 3.76 (s, 3H, -OCH_3_), 4.81 (s, 2H, NH_2_), 4.91 (s, 2H, CH_2_), 6.84 (dd, 1H, *J* = 1.5, 6.5, Har), 7.19 (d, 2H, *J* = 8.5 Hz, Har), 7.23 (d, 2H, *J* = 8 Hz, Har), 7.38 (m, 3H, Har), 11.36 (s, 1H, NH-indole). ^13^C NMR (175 MHz, DMSO-*d*_6_) *ppm*: 21.2 (-CH_3_), 54.3 (CH_2_), 55.7 (OCH_3_), 102.3, 108.9, 113.7, 115.0, 127.5, 127.6, 128.6, 129.3, 129.6, 131.2, 131.7, 134.2, 136.9, 154.0, 162.6 (C=O); MS *m*/*z* (ESI): 308 [M − H]^+^; anal. calcd. for C_18_H_19_N_3_O_2_: C, 69.88; H, 6.19; N, 13.58; found: C, 70.04; H, 6.17; N, 13.54.

### 3.3. MTT Assay

Anti-proliferative activity in compounds **4a**–**q** were tested in three cancerous cell lines (MCF-7 breast cancer cells, A549 lung carcinoma, HCT human colon cancer cells) and in WI-38 human lung fibroblast cells. Cells were plated at a density of 1.2–1.8 × 10,000 cells/well in a volume of 100 µL complete growth medium and were incubated at 37 °C for 24 h. Cells were then treated with serially diluted tested compound and incubated at 37 °C for 48 h. Twenty microliters of 3-(4,5-dimethylthiazol-2-yl)-2,5-diphenyltetrazolium bromide (MTT) (2.5 mg/mL) dissolved in PBS were added to the cells and cells were further incubated for 4 h at 37 °C. To dissolve formazan crystals, MTT solutions were completely removed and 100 μL DMSO was added. The absorbance was detected at 540 nm using a Spectramax 250 microplate reader (Molecular device, San Jose, CA, USA), and viability (%) was calculated as [optical density (OD) of treated group/OD of control group] × 100.

### 3.4. Flow Cytometry

To detect the apoptotic potential of our compound of choice, **4e**, an Annexin V-FITC Apoptosis Detection Kit (BioVision, Mountain View, CA, USA) was used based on the manufacturer’s protocol using a dose of 10 μm for 48 h.

### 3.5. Cell Cycle Arrest

To detect cell cycle status after treating cells with **4e**, 1 × 10^6^ cells were cultured in six-well plates for twenty-four hours per cell and were then treated with 10 μm of **4e** for 48 h. Cells were then washed with phosphate-buffered saline. Cells were then fixed with 70% of cold ethanol and kept at 4 °C overnight. Cell analysis was then analyzed following the manifacorere protocol, using a Propidium Iodide Flow Cytometry Kit for Cell Cycle Analysis (Abcam, Waltham, MA, USA).

## 4. Conclusions

Herein, we reported the design and synthesis of a series of substituted-*N*-benzyl-1*H*-indole-2-carbohydrazide **4a**–**q**. All the synthesized compounds were tested for their antiproliferative activity against three cancer cell lines, namely, MCF-7, A549, and HCT using an MTT assay. Several compounds showed moderate to high cytotoxicities with IC50, similar or superior to the reference drug Staurosporine. In particular, compound **4e** was the most active congener from the series, with IC50s as low as 0.57, 1.95, and 3.49 µM, respectively, against MCF7, HCT116, and A549 compared to the IC50s of Staurosporine (11.1, 7.02, and 8.42 µM, respectively). Additionally, the selectivity of all the series were evaluated against a non-tumorigenic cell line (WI38), and compound **4e** exhibited selectivity toward cancer cells. Finally, flow cytometry suggested that compound **4e** harbors potential apoptosis-inducing capabilities. Based on the findings, compound **4e** seems to be a promising lead compound for further investigation.

## Data Availability

Data are contained within the article or Appendix A.

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
