# Peer review of "A Facile Synthesis and Molecular Characterization of Certain New Anti-Proliferative Indole-Based Chemical Entities"

_ijms, 2023, doi:10.3390/ijms24097862_

Round 1

Reviewer 1 Report

The present work presents the experimental results regarding the synthesis of new indole-based organic compounds and their cytotoxic activity against three different cell lines. The paper needs to be improved and the compound characterisation requires experiments to determine the real compound structure.

I have some suggestion:

Line 2: I would put certain away.

Line 19: I would not mention the worst results on the abstract.

Line 21: I would invert the order: showed from moderate to excellent...

Line 28: was instead of were.

Lines 39-41: Is it related to indole? It is not clear the purpose of the sentence.

Lines 70-71: The amine displace the leaving group through an addition-elimination mechanism. It is not a nucleophilic substitution.

Line 76: other instead of another.

Scheme 1: Why is R1 always an electron withdrawing substituent (but for 4q)? 

Line 188: What are the yields?

Regarding the 4a-q structure, did you find any literature references confirming that it is the most hindered nitrogen that attacks the carbonyl group?

Did you run any 1H-NMR spectra with D2O to confirm which peaks relate to NH bond?

Did you run any N-NMR spectra to confirm the structure?

Line 198: only one signal is observed for NH2. How do you explain that considering that NH2 have asymmetrical stretching movements?

The 4a-q structure is not unequivocally determined.

Author Response

  • The present work presents the experimental results regarding the synthesis of new indole-based organic compounds and their cytotoxic activity against three different cell lines. The paper needs to be improved and the compound characterisation requires experiments to determine the real compound structure.

First I would like to thank you for your valuable comments, it pointed out many things in the paper. The manuscript has been improved and the characterization of the compound was confirmed with x-ray crystallography and D2O NMR a copy is attached.

  • Line 2: I would put certain away; certain can be removed.
  • Line 19: I would not mention the worst results on the abstract; the abstract has been rconstructed.
  • Line 21: I would invert the order: showed from moderate to excellent; the order was converted to moderate to high.
  • Line 28: was instead of were; corrected.
  • Lines 39-41: Is it related to indole? It is not clear the purpose of the sentence; yes it contains indole ring and the sentence was connected with also and become more clear.
  • Lines 70-71: The amine displace the leaving group through an addition-elimination mechanism. It is not a nucleophilic substitution; That is true and has been corrected.
  • Line 76: other instead of another; done.
  • Scheme 1: Why is R1 always an electron withdrawing substituent (but for 4q)?: Well we working on similar series of hydrazones and we noticed that the electron withdrawing groups have better biological effects.
  • Line 188; What are the yields?; the yields has been added.

Regarding the 4a-q structure, did you find any literature references confirming that it is the most hindered nitrogen that attacks the carbonyl group?; Even though we confirmed the structure by x-ray crystallography and we going to publish the data elsewhere, the D2O also confirm the disappearance of the NH2 signal.  The explanation may be due to as Wang et al. hypothesized that the terminal NH2 of the phenylhydrazines was blocked under the reaction conditions by the agent he used. Furthermore, the reactivity of the secondary amine was enhanced by the α-effect as suggested by Nigst, et al.

  • Wang, X. Di, C. Wang, L. Zhou, J. Sun, Org. Lett.201618, 1900-1903.
  • A. Nigst, A. Antipova, and H. May; Nucleophilic Reactivities of Hydrazines and Amines: The Futile Search for the α-Effect in Hydrazine Reactivities. J. Org. Chem.2012, 77, 18, 8142–8155.
  • Did you run any 1H-NMR spectra with D2O to confirm which peaks relate to NH bond?; We did the D2O NMR and the two proton signal disappeared. A copy is attached.
  • Did you run any N-NMR spectra to confirm the structure? Unfortunately it was not run but we did x-ray crystallography and it confirmed the structure. A copy is attached.
  • Line 198: only one signal is observed for NH2. How do you explain that considering that NH2have asymmetrical stretching movements? Was added.
  • The 4a-q structure is not unequivocally determined; I think with the presence of the two proton in the 1HNMR and it disappearance of it in the D2O spectra and they are not shown in the 13CNMR added to that the x-ray crystallography we think it is equivocally determined.
  • The present work presents the experimental results regarding the synthesis of new indole-based organic compounds and their cytotoxic activity against three different cell lines. The paper needs to be improved and the compound characterisation requires experiments to determine the real compound structure.

Reviewer 2 Report

The paper entitled “A Facile Synthesis and Molecular Characterization of Certain New Anti-proliferative Indole-Based Chemical Entities” describes the synthesis and biological evaluation of novel indole-based amides showing excellent to moderate cytotoxicity with potential anticancer activity. Specifically, compound 4e demonstrated to be the most active molecule with IC50 = 2 uM. Biological results are notheworthy even if the paper is considered routine medicinal chemistry. No significant improvement in chemical protocols is reported. Moreover, several revisions in the chemistry section are mandatory before publication in IJMS.

The author stated that "the higher nucleophilicity of the secondary amine in compounds 3 than its primary amine led to formation of the sole product in the target compounds 4a-q." This is not correct, as you can see hydrazines usually react as ambident nucleophiles depending on reaction conditions (https://pubs.acs.org/doi/pdf/10.1021/ja00459a036), while form hydrazones with aldehyde in position N’. Could you please provide a reference of this transformation involving the use of coupling reagents? Otherwise a bidimensional NMR experiment could be useful to confirm the NH2 signal.

Chemical procedures are poorly described. First, please provide correctly g, mol and equiv for the protocols and grams of the final compounds obtained next to the yield. 

The author performed MS m/z (ESI) analysis to confirm the structures. Example: 297.8 [M - H]+, please check carefully if it is positive or negative ionized result!

The authors need to check carbon signals: a lot of carbons are missing! A complete list of the carbon signals need to be included, because in several compounds (ex 4a, 4b, 4c, 4g, 4h...) many signals are not reported. Need to explicit everytime MHz of the instruments and spectroscopic data need to be reported as follow (according to Journal's requirements): 1H NMR (500 MHz, DMSO-d6): δ  13C NMR (125 MHz, DMSO-d6): δ…

For compounds 4e and 4f: are you able to identify the CF3 signals in the spectra?

Copies of 1H and 13C NMR for all compounds need to be included in the supplementary materials to assure structural information and purity >90%. Also provide a single PDF files with all the spectra.

Other minor points:

40: modify panbinostat with Panobinostat

66: EDCI hydrochloride?

160: modify DMSO-d6 with DMSO-d6

161: change mole with mol

207: there is no assignment for signals 4.84 and 5.04, please furnish it

Author Response

  • The paper entitled “A Facile Synthesis and Molecular Characterization of Certain New Anti-proliferative Indole-Based Chemical Entities” describes the synthesis and biological evaluation of novel indole-based amides showing excellent to moderate cytotoxicity with potential anticancer activity. Specifically, compound 4e demonstrated to be the most active molecule with IC50 = 2 uM. Biological results are noteworthy even if the paper is considered routine medicinal chemistry. No significant improvement in chemical protocols is reported. Moreover, several revisions in the chemistry section are mandatory before publication in IJMS; We would like to thank the reviewer for his valuable comments which improved the manuscript and highlighted important points. Even though the chemistry of the scheme maybe not novel but we think our series have good potential as anticancer candidate especially compound 4e as you mentioned had showed an IC50 of 0.57 µM in MCF-7 and average IC50 of 2 µM. Furthermore, it was found that it is more selective against cancerous cell line than normal cell line.    
  • The author stated that "the higher nucleophilicity of the secondary amine in compounds 3 than its primary amine led to formation of the sole product in the target compounds 4a-q." This is not correct, as you can see hydrazines usually react as ambident nucleophiles depending on reaction conditions (https://pubs.acs.org/doi/pdf/10.1021/ja00459a036), while form hydrazones with aldehyde in position N’. Could you please provide a reference of this transformation involving the use of coupling reagents? Otherwise a bidimensional NMR experiment could be useful to confirm the NH2 signal;

I agree that with the reference you provided, ambient reaction conditions as the one we used may aid the alkylation to be on the secondary amine. Also, this may be attributed as wang et al. hypothesized that the terminal NH2 of the phenylhydrazines was blocked under the reaction conditions by agent he used. Furthermore, the reactivity of the secondary amine was enhanced by the α-effect as suggested by Nigst, et al.

The compounds were confirmed by x-ray crystallography and D2O NMR a copy is attached. Also, some paper confirms the  

  • Wang, X. Di, C. Wang, L. Zhou, J. Sun, Org. Lett.201618, 1900-1903.
  • A. Nigst, A. Antipova, and H. May; Nucleophilic Reactivities of Hydrazines and Amines: The Futile Search for the α-Effect in Hydrazine Reactivities. J. Org. Chem.2012, 77, 18, 8142–8155.
  • Chemical procedures are poorly described. First, please provide correctly g, mol and equiv for the protocols and grams of the final compounds obtained next to the yield; chemical procedure is now revised and the information needed are included.
  • The author performed MS m/z (ESI) analysis to confirm the structures. Example: 297.8 [M - H]+, please check carefully if it is positive or negative ionized result!; It was in the negative mode.
  • The authors need to check carbon signals: a lot of carbons are missing! A complete list of the carbon signals need to be included, because in several compounds (ex 4a, 4b, 4c, 4g, 4h...) many signals are not reported; the carbon signals were checked and the missing carbons were added, but some carbons may have the same chemical environment and appear in the same chemical shift that may explain some of the missing carbons.
  • Need to explicit everytime MHz of the instruments and spectroscopic data need to be reported as follow (according to Journal's requirements): 1H NMR (500 MHz, DMSO-d6):δ … 13C NMR (125 MHz, DMSO-d6): δ; were added according to the journal requirements.
  • For compounds 4e and 4f: are you able to identify the CF3 signals in the spectra?; Yes the splitting of the CF3 signals was identified on the FID files as a quartet signal.
  • Copies of 1H and 13C NMR for all compounds need to be included in the supplementary materials to assure structural information and purity >90%. Also provide a single PDF files with all the spectra; copies are provided.
  • 40: modify panbinostat with Panobinostat; done.
  • 66: EDCI hydrochloride?; yes it is EDCI.HCl and it was corrected in the manuscript.
  • 160: modify DMSO-d6 with DMSO-d6; done.
  • 161: change mole with mol; done.
  • 207: there is no assignment for signals 4.84 and 5.04, please furnish it; Done, sorry for the mistake.

Round 2

Reviewer 1 Report

The paper was improved according to the suggested modifications.

It can be published.

Reviewer 2 Report

After the suggested revision I consider now the paper for acceptance!

Good luck